# Canadian adolescents' perceptions of how climate change is impacting their mental health: A qualitative analysis of open-ended survey responses

Ishwar Tiwari[1], Rebekah Alice McKinnon[1], Azeezah Jafry[1], Ekroop Grewal[2], Jason Gilliland[3,4,5,6], Kendra Nelson Ferguson[3], Kiffer G. Card[7], Maya Gislason[7], Alina Paula Cosma[8], Gina Martin[1,3]*

1 Faculty of Health Disciplines, Athabasca University, Athabasca, Canada, 2 Emerging Youth Consultancy, Toronto, Canada, 3 Department of Geography & Environment, Western University, London, Canada, 4 Western University, Human Environments Analysis Lab, London, Canada, 5 Department of Epidemiology & Biostatistics, Western University, London, Canada, 6 Department of Paediatrics, School of Health Studies, Western University, London, Canada, 7 Faculty of Health Sciences, University Drive, Simon Fraser University, Burnaby, Canada, 8 School of Psychology, Trinity College Dublin, Dublin, Ireland

* gmartin@athabascau.ca

## Abstract

There is increasing recognition that climate change affects mental health, with young people identified as a high-risk population. Yet, research on this topic has mostly focused on quantitative studies with limited research exploring youth's perspectives. This study explored Canadian adolescents' perceptions of the impacts of climate change on their mental health. Data were collected from a cross-sectional survey of 804 Canadian adolescents (13–18 years). This study utilized open-ended response data from participants who reported that their mental health was impacted by climate change and who were then asked to describe how. An inductive thematic analysis with semantic coding was undertaken to analyze the data. Thirty-seven percent of participants reported that they felt climate change was impacting their mental health either a little or a lot, and 235 participants answered the open-ended question about how. In the open-ended responses, four themes emerged: i) Emotional and psychological responses, ii) Concerns for the future, iii) Impacts on functioning, and iv) Concerns for the environment, humanity, and wildlife. This study highlights that adolescents' perspectives of the impacts of climate change on their mental health were diverse, highlighting multiple pathways linking climate change to mental health among this demographic. These insights can inform strategies to mitigate the climate-related mental health impacts among youth.

**Data availability statement:** The dataset is not available for sharing publicly because we do not have ethical approval to share beyond our immediate research team. The Research Ethics Board who has imposed this restriction is Athabasca University Research Ethics Board (contact: rebsec@athabascau.ca). Anonymized data may be made available, upon reasonable request, to the corresponding author and following ethical approval.

**Funding:** This study was supported by Social Sciences and Humanities Research Council (Grant # 430-2022-00362 to GM). The funders had no role in study design, data collection and analysis, decision to publish, or preparation of the manuscript.

**Competing interests:** The authors have declared that no competing interests exist.

# 1. Introduction

The global climate is changing rapidly; this includes rising temperatures, altered precipitation patterns, and, with this, an increase in the frequency, duration, and severity of extreme weather events [1]. These trends are evident in Canada, the second-largest country in the world by land mass, which experiences diverse and often extreme climatic conditions [2]. Over the past decade, Canada has experienced many extreme weather and related events that have garnered significant media attention, including heatwaves and wildfires [3,4]. These changes in climate and related events are predicted to intensify over the coming decades [1], to the detriment of human health [5]. Systematic reviews have reported adverse effects on human health as a result of climate change, including increased instances of infectious diseases and neurological conditions, as well as deterioration of cardiorespiratory health [6–9]. In recent decades, the connection between climate change and mental health has been highlighted as a growing area of concern recognized by numerous key organizations, including the American Psychological Association, the American Psychiatric Association, the World Health Organization, and the United Nations Children's Fund [10–12].

Climate change can impact mental health through multiple (not mutually exclusive) pathways. Direct effects can occur after experiencing an acute event in one's community, such as extreme weather or a related event (e.g., wildfires, floods, and heatwaves). This can cause physical harm, displacement, and destruction of property, leading to acute stress and trauma [13]. The indirect effects of climate change can occur through its impact on various socioeconomic, political, and environmental factors that influence mental health [14]. Further, overarching effects can occur due to the psychological distress that arises from awareness of climate change and the threat it poses to the Earth's systems [15–17].

Understanding the impacts of climate change on key sub-populations is imperative for effectively mitigating negative mental health outcomes. Adolescents are a sub-population undergoing significant physical, mental, and social development [18]. As a result, they are considered more susceptible to the mental health impacts of climate change [19,20]. However, much of the existing literature has focused on adult populations [21–26]. A commentary article that is based on existing evidence has argued that extreme weather and related events, climate-induced forced migration (e.g., displacement due to wildfires), and the anticipation of future hazards (e.g., fear of food insecurity, increased frequency and severity of extreme weather events) may lead to psychological distress and have prolonged effects on adolescents' mental health [27].

Existing studies exploring the relationship between climate change and mental health in young people have primarily employed quantitative methods [28–31]. For example, Hickman et al. [28] employed eight closed-ended questions that focused on several domains, including participants' climate-related emotions, which consisted of 14 positive and negative emotions related to climate change. Further, evidence has shown that young people perceive that climate change is impacting their mental

health. Galway and Field [32] used a closed-ended survey question to determine the degree to which the feelings Canadian youth aged 16–25 have about climate change influences their mental health. They found that over three-quarters of the participants reported some degree of impact on their mental health; additionally, more than half (56%) reported feeling scared, sad, anxious, and powerless [32]. Although these quantitative approaches have provided valuable insight into the various emotions adolescents are feeling about climate change and whether they feel climate change is impacting their mental health, they do not allow for a more in-depth exploration of the nuances of the subjective experiences of adolescents through unstructured responses in their own words, specifically, how adolescents perceive that climate change is impacting their mental health. In contrast to quantitative approaches, which can measure the phenomena under study, qualitative research can provide useful insights into the 'how, especially when evidence is limited [33]. Employing open-ended questions allows for person centered approaches to explore adolescents' perspectives and lived experiences [34], offering a deeper understanding of the complex interplay of the relationship between climate change and mental health and allowing for unanticipated responses. Additionally, while a small but growing body of qualitative research has explored youth climate-related mental health impacts in other parts of the world (e.g., UK, USA, France) [35,36], studies centering Canadian adolescents' perspectives, particularly in capturing the nuanced nature of mental health impacts, are limited.

To address the aforementioned gap, this study uses close and open-ended questions to shed light on the research questions: i) To what degree Canadian adolescents perceive that climate change is impacting their mental health? and ii) How do they describe the impacts of climate change on their mental health? The research questions and approach align with the theorized pathways (direct, indirect, overarching) that have been proposed about how climate change may impact mental health, as it allows the participants to answer in their own words how/if they are experiencing these phenomena, rather than limiting them to questions on one or two specific pathways.

## 2. Methods

### 2.1. Ethics statement

The study procedures were approved by the Athabasca University Research Ethics Board (#25013). Before undertaking the survey, potential participants were provided with a Letter of Information outlining the aims, length, and content of the survey. Potential participants were assured of the anonymity and confidentiality of their data, as well as the voluntary nature of their participation. We obtained written consent from the participants before they proceeded with the survey questions. Participants received compensation for their time, as per the terms they agreed to when joining the panel, for example, participants could elect to get points which they could redeem with specific merchants.

### 2.2. Research design

From October to November 2023, we conducted a cross-sectional web-based survey to explore adolescents' thoughts, feelings, beliefs, and experiences regarding climate change. To be eligible to complete the survey, participants had to live in Canada and be aged 13–18.

Adolescents were recruited to participate in the survey through a pre-registered online research panel through Qualtrics [37]. Potential participants were targeted based on core demographic information held by Qualtrics about the panel members; Qualtrics conducted the recruitment. Participants under the age of 18 were required by Qualtrics to have parental/guardian consent to be part of the research panel and a parent/guardian was required to approve that any potential participants could be recruited into this study. We adopted a non-probability quota-based approach [38] to recruit at least 100 participants from each of eight regions across Canada: Alberta, British Columbia, Saskatchewan and Manitoba, Ontario, Quebec, the Atlantic provinces (Newfoundland and Labrador, New Brunswick, Nova Scotia, and Prince Edward Island), and the Northern Territories (i.e., Northwest Territories, Yukon, and Nunavut) with a target quota of 800 participants. The quota was met in all regions except the Northern Territories. We aimed to recruit equal numbers of boys and girls (300

boys and 300 girls) while leaving space for gender-diverse young people. We also aimed to recruit participants in equal numbers from two age classifications: 400 'younger' adolescents aged 13–15 and 400 'older' adolescents aged 16–18, as younger adolescents tend to be underrepresented in studies about climate change and mental health [39], and at least 300 participants residing in non-urban areas. All age, gender, and urbanicity quotas were met.

In total, 1542 potential participants accessed the link and answered if they agreed to be part of the study, with 87.22% (n = 1345) stating they agreed. Screening questions based on the quota characteristics and eligibility were then asked, leaving 895 eligible participants. Qualtrics and the research team performed a data quality check to identify and exclude poor-quality data, such as duplicates, click throughs, or straight lining, from the surveyed participants, which left 804 participants who were included in the study.

## 2.3. Survey instrument and data collection

The full survey instrument was created and refined through focus groups that examined questions used in previous studies used to measure climate emotions and coping strategies among young people (seven online focus groups, participants aged 15–18) as well as by an interdisciplinary team comprised of academics (e.g., geography, health disciplines, and psychology) and three high school aged youth co-researchers. After developing the survey, we piloted the questions about climate change through cognitive interviews (seven participants, aged 13–18) and adjusted accordingly to ensure these key questions were comprehensible and had face validity [40].

In this study, we analyzed data from participants who responded "yes, a little" or "yes, a lot" to the question, "Do you think climate change is impacting your mental health?" (Response options were: "no," "yes, a little," and "yes, a lot"). These participants were then asked to respond to an open-ended question, where they could fill in an open text box. The prompt to the question was: "Based on your experience, how is climate change impacting your mental health?"

## 2.4. Data analysis

Of 804 participants, 17 participants were not asked this question because they indicated earlier that they did not believe climate change was happening, when asked if they agreed with a statement that climate change was mainly due to human activity (response options: "yes", "no", "I don't know", "I don't believe climate change is happening"). Those who did not believe climate change is happening were not asked the suite of questions about thoughts and feelings about climate change as these may feel irrelevant to these participants or be interpreted in a different way than the rest of the sample. Of the remaining participants, 776 completed the closed-ended question of whether climate change was impacting their mental health. Four hundred ninety-three participants (63%) reported that climate change was not impacting their mental health, while 291 (37%) reported that it was impacting their mental health, either 'a little' (n = 213) or 'a lot' (n = 78). Of these 291 participants, a total of 235 open-ended responses were utilized for analysis; 56 responses were excluded based on predefined exclusion criteria: (1) participants provided responses that did not address the question prompt or were too vague for further analysis (e.g., "I don't know," "mentally," "negatively"; n = 42) and (2) responses were missing entirely (n = 14).

We analyzed the open-ended responses using an inductive thematic analysis approach [41], guided by a post-positivist paradigm [42]. This approach positions that participants' responses reflected real perceptions and experiences while recognizing that interpreting them requires structured methods with the aim to reduce researcher bias and stay close to the participants' intended meaning. We used semantic coding, as well as double-coding with consensus-building, and incorporated code frequencies (to help assess patterns across the large and diverse dataset [41].

An experienced qualitative researcher, A.J., led the coding process, assisted by a youth co-researcher, E.G., who served as the second coder. E.G. was trained by A.J. in semantic coding and Dedoose [43], a qualitative and mixed-methods data analysis software, which was used to review, organize, and combine codes. Both coders worked

independently to familiarize themselves with the data and to generate initial codes and subcodes. To ensure consistency, the two coders met regularly to compare their coding and resolve discrepancies through discussion until they reached a consensus. In this way, they acted as "critical friends" by helping to explore alternative perspectives of the data [44]. Once the initial codes were generated, a third researcher (I.T.) grouped codes into preliminary themes to capture recurring ideas or patterns in the data, which the other authors reviewed. Any discrepancies were discussed between researchers, leading to themes being altered and refined. Selected exemplary quotes are presented in the results, along with the age, gender, urbanicity of area of residence, and province of the participants for context. Some quotes contain the full responses for each theme/subtheme for flow and context, while others include only the relevant parts of those responses.

## 3. Results

The socio-demographic characteristics of the overall participants and the subset whose responses to the open-ended question were analyzed are presented in Table 1. Among the analytical subset, the mean age was approximately 16, and nearly 55% identified only as a girl (i.e., cis girl), with a small proportion (3.4%) representing diverse gender identities (i.e., non-binary, transboy, transgirl, two-spirit, or gender fluid). Geographically, more than half of the participants (68.1%) resided in urban areas. Furthermore, 61% of the participants were White, and approximately half (46.8%) perceived their family's financial position as average. Regarding perceived mental health impacts of climate change, over three-quarters of the analytical subset of participants reported their mental health was impacted a little (75.7%), while nearly a quarter (24.3%) reported a lot.

### 3.1. Thematic results

The responses of 235 participants held 382 codes, as a single open-ended response could contain multiple codes representing the mental health impacts experienced by the participants due to climate change. The coding process identified 85 subcodes which were grouped into 42 parent codes provided as a supplementary materials (S1 and S2 Tables).

From the 42 codes representing similar concepts, four main themes were identified, with some themes having more specific subthemes within them (Table 2): i) emotional and psychological responses (subthemes: i.i general affective state, i.ii emotional or psychological responses connected to climate change-related extreme events and related conditions, i.iii exacerbation of existing mental health concerns, i.iv emotional and psychological responses to perceived systemic barriers to climate action, and i.v negative emotions about climate change connected to climate change information), ii) concerns for the future (subtheme: ii.i uncertainty about the future and ii.ii concerns about parenthood), iii) impacts on functioning (subthemes: iii.i impact on daily functioning and iii.ii physiological impacts), and iv) concerns for environment, humanity, and wildlife. Some subthemes were less frequently coded; however, we retained them as standalone categories to uncover nuanced perspectives, thus maintaining consistency with our inductive, participant centered approach.

#### i. Emotional and Psychological Responses

The theme of emotional and psychological responses was the dominant theme. It accounted for 229 coded responses (59.9% of total codes), representing participants' emotional and psychological responses to climate change. These included responses that stated experiences of anxiety, stress, worry, depression, sadness, and fear. This theme comprised five subthemes: general affective state, emotional or psychological responses to specific weather conditions and related events, exacerbation of existing mental health concerns, emotional and psychological responses specific to systemic barriers to climate action, and negative emotions about climate change connected to climate change information.

#### i.i. General Affective State

There were 17 unique codes describing the participants' general negative emotional state, accounting for 133 coded responses (34.8%). This subtheme captured the participants' feelings of being distressed by climate change and its

**Table 1. Socio-demographic characteristics of the overall participants (N=804) and of the subset of study participants whose open-ended responses were analyzed (N=235).**

| Sample characteristic | Full Sample (n=804)<br>n (%)/ Mean (SD), Min, Max | Subset with Open-ended Responses (n=235)<br>n (%)/ Mean (SD), Min, Max |
|---|---|---|
| Age | 15.6 ((± 1.65), 13, 18 | 15.7 (± 1.71), 13, 18 |
| Province | | |
| Alberta | 123 (15.0%) | 42 (17.9%) |
| British Columbia | 123 (15.0%) | 35 (14.9%) |
| Manitoba | 70 (8.7%) | 19 (8.1%) |
| New Brunswick | 48 (6.0%) | 15 (6.4%) |
| Newfoundland and Labrador | 26 (3.2%) | 8 (3.4%) |
| Northern Territories (Northwest Territories, Yukon, and Nunavut) | 28 (3.5%) | 6 (2.5%) |
| Nova Scotia | 52 (6.5%) | 13 (5.5%) |
| Ontario | 135 (17.0%) | 41 (17.4%) |
| Prince Edward Island | 2 (0.2%) | – |
| Quebec | 128 (16.0%) | 35 (14.9%) |
| Saskatchewan | 69 (8.6%) | 21 (8.9%) |
| Urban/rurality | | |
| A city | 550 (68.4%) | 160 (68.1%) |
| A rural or remote area | 65 (8.1%) | 20 (8.5%) |
| A small town | 187 (23.3%) | 54 (23.0%) |
| Unknown | 2 (0.2%) | 1 (0.4%) |
| Gender | | |
| Girl | 408 (50.7%) | 129 (54.9%) |
| Boy | 371 (46.1%) | 98 (41.7%) |
| Gender diverse | 25 (3.1%) | 8 (3.4%) |
| Race/ethnicity | | |
| White | 460 (57.2%) | 144 (61.3%) |
| Black | 108 (13.4%) | 29 (12.3%) |
| South Asian | 48 (6.0%) | 18 (7.7%) |
| Indigenous (First Nation, Inuit/Inuk, Metis) | 21 (2.6%) | 11 (4.7%) |
| Other race or ethnicity (including multiple selections) | 167 (20.8%) | 33 (14.0%) |
| Perceived Family Affluence | | |
| Not well-off | 118 (14.6%) | 49 (20.9%) |
| Average | 385 (48.0%) | 110 (46.8%) |
| Well-off | 301 (37.4%) | 56 (32.3%) |
| Climate change is impacting mental health | | |
| Unknown | 513 (63.8%) | – |
| Yes, a little | 212 (26.4%) | 178 (75.7%) |
| Yes, a lot | 79 (9.8%) | 57 (24.3%) |

Max: Maximum.

Min: Minimum.

SD: Standard Deviation.

**Table 2.** Number of coded responses within themes and subthemes identified through the thematic analysis.

| Theme (number of codes, n = 42) | Subtheme (number of codes, n = 42) | Number of Coded Responses (n = 382) |
|---|---|---|
| i.Emotional and psychological responses 22(25) | i.i General affective state (17) | 133 |
| | i.ii Emotional or psychological responses connected to climate change-related extreme events and related conditions(2) | 57 |
| | i.iii Exacerbation of existing mental health concerns(2) | 27 |
| | i.iv Emotional and psychological responses to perceived systemic barriers to climate action(2) | 7 |
| | i.v Negative emotions about climate change connected to climate change information(2) | 5 |
| ii.Concerns for the future(6) | ii.i Uncertainty about the future(5) | 90 |
| | ii.ii Concerns about parenthood(1) | 4 |
| iii.Impacts on functioning(8) | iii.i Impact on daily functioning(7) | 16 |
| | iii.ii Physiological impacts(1) | 9 |
| iv.Concerns for the environment, humanity, and wildlife(3) | Not applicable (see theme) | 34 |

consequences. For example, one participant stated, "*climate change is making me feel depressed*" (Participant # 9, girl, age 17, city, Alberta). Another participant shared, "*climate change caused me a lot of anxiety to think about it*" (Participant # 57, girl, age 17, city, Alberta). Some participants also focused on the feeling of powerlessness. For example, one participant stated, "*climate change gives me anxiety and I feel powerless*" (Participant # 640, boy, age 13, small town, Nova Scotia).

### i.ii. **Emotional or Psychological Responses Connected to Climate Change-Related Extreme Events and Related Conditions**

This subtheme accounted for 57 coded responses (14.9%), capturing participants' emotional responses, in which they connected their emotional and psychological states to a specific extreme event or weather condition. Some participants highlighted the impact that wildfires were having on their mental health. For example, one participant stated, "*all the wildfires are giving me anxiety*" (Participant # 588, boy, age 15, rural area, British Columbia). Another participant echoed this, stating, "*it [climate change] is impacting my mental health as I am worried about wildfires*" (Participant # 533, boy, age 13, city, British Columbia). Some expressed a fear of direct exposure to wildfire. For instance, one participant stated, "*in the summer, I am very scared that there will be a forest fire near my house*" (Participant #768, girl, age 14, small town, British Columbia). Other participants noted that heatwaves were impacting their mental health. One participant shared, "*the extreme heatwaves and disasters linked to climate change have seriously affected my mental well-being, making me feel scared and uncomfortable*" (Participant # 679, girl, age 14, rural area, Manitoba). Worry about heat was also expressed by another participant, who was also concerned about storms and their potential impact on society. The participant shared, "*I worry that the world is going to get too warm and we are going to have more storms and problems with power and access to technology*" (Participant # 507, boy, age 13, city, New Brunswick). Some participants expressed that more day-to-day weather conditions, rather than climate change-related extreme events, influence their emotional state. For example, one participant shared the impact of extreme temperatures on their mood, stating, "*the weather has affected me. It being too hot or too cold affects my mood*" (Participant # 451, boy, age 16, city, British Columbia).

### i.iii. **Exacerbation of Existing Mental Health Concerns**

Participants' perspectives on how climate change exacerbates pre-existing mental health problems emerged in 27 coded responses (7.1%). Some participants shared how environmental conditions influence their mental state. For

example, one participant stated, "*I have had severe depression since the age of 12 and a borderline personality disorder, and climate change plays a huge role in my emotions*" (Participant # 348, girl, age 17, small town, Quebec). Other participants expressed how the burden of thinking about climate change amplifies their negative mental states. For instance, one participant shared, "*I'm already feeling a bit lethargic/depressed most of the time. Worrying about climate change is just something that adds to it*" (Participant # 181, girl, age 14, city, Alberta).

### i.iv. Emotional and Psychological Responses to Perceived Systemic Barriers to Climate Action

The subtheme of responses to perceived systemic barriers to climate action consisted of 2 codes and accounted for 7 coded responses (1.8%), representing participants' emotional and psychological responses specific to perceived barriers to climate change mitigation efforts. Overall, participants expressed negative emotions due to those in positions of authority and influence who they perceived as not prioritizing environmental sustainability. For example, one participant stated, "*it makes me sad to know that big corporations that produce loads of carbon dioxide would rather have a lot of money than a healthy planet*" (Participant # 46, girl, age 18, city, New Brunswick). Some participants also expressed their anger towards people in power for their apparent lack of concern for younger generations. A participant stated, "*I sometimes get angry thinking of the people in powerful positions not caring about our future*" (Participant # 238, boy, age 18, city, British Columbia).

### i.v. Negative Emotions about Climate Change Connected to Climate Change Information

The impacts of information about climate change on mental health appeared in 5 coded responses (1.3%). Participants expressed that the continuous flow of information about the negative impacts of climate change has a negative impact on their mental health. One participant stated, "*constant reminders of the negative effects of climate change have been slightly impacting my mental health because of how serious and large this problem is*" (Participant #217, girl, age 14, city, Ontario). Another participant echoed this sentiment, stating, "*I often find myself stressed and anxious because of the non-stop flow of news about climate change*" (Participant #709, girl, age 14, small town, Northwest Territories). The role of media exposure in amplifying this stress was highlighted by another participant, who stated they were "*constantly worrying about what the news outlets are saying about the world*" (Participant #514, girl, age 18, city, Alberta).

### ii. Concerns for the Future

This theme is the second most dominant theme and accounted for 94 coded responses (24.6%). Participants' perspectives about the short- and long-term influences of climate change on their future, their loved ones, and future generations are captured by this theme. The theme encompasses two subthemes: uncertainty about the future and concerns about parenthood.

### ii.i. Uncertainty about the Future

This subtheme consisted of 5 codes and accounted for 90 coded responses (23.6%) that captured adolescents' expression of an uncertain future due to climate change. Feelings of worry and hopelessness were tied to uncertainty about the future. One participant shared, "*I am scared about what will happen in the future, and it keeps worrying me*" (Participant # 294, girl, age 15, city, Alberta). For some, their sense of a positive future for themselves has been impacted by climate change. One participant expressed, "c*limate change has often made me feel as though anything I want to achieve far in the future doesn't have a purpose because climate change is very close to reaching a point of no return*" (Participant # 166, girl, age 15, city, Alberta). Another participant echoed this concern, stating that climate change "*dwindles my hope for the future and what I'll be able to do with my life if the world is destroyed*" (Participant # 116, gender diverse, age 18, city, Alberta). Another participant shared, "*I get depressed thinking about all the things I might lose to climate change*" (Participant # 346, gender diverse, age 17, small town, Saskatchewan).

Participants also shared specific concerns about the potential future loss of homes and livelihoods due to climate change-related extreme weather events. One participant explained:

> *It's making me feel anxious about the future, not just for myself but for my family, my friends and future generations. A lot of my family live in areas that will be impacted severely by climate change (coastal regions), and it's worrying thinking about how they might lose their homes due to flooding or land loss* (Participant # 50, girl, age 18, city, Nova Scotia).

Some participants shared their fear for future generations and connected this to a sense of lack of caring and empathy of those in the present. One participant stated, "*the thought of the fact that the earth we live on is dying, and no one cares, makes me fear for the future generations*" (Participant # 728, girl, age 14, rural area, Ontario). Others expressed concern about people's ability to address climate change and a sense of pointlessness given a dismal future. As one participant stated, "*it's a worry that's always on the back of my mind. When I think about it, I feel like there's no point in anything because we're going to become extinct*" (Participant # 23, girl, age 18, city, Ontario).

### ii.ii. Concerns about Parenthood

Concerns about becoming a parent due to the worsening environmental conditions and the challenges future generations may face appeared in 4 coded responses (1.0%). One participant shared that the decision whether or not to have children weighed heavily on them due to climate change:

> *It's depressing to think about the future and whether or not I should have kids, depending on the world situation. I've always wanted to be a mother, but if things continue to go the way they are, I don't know if I should* (Participant # 117, girl, age 17, city, British Columbia).

Another participant expressed similar worries, stating, "*it scares me and stresses me out; I'm worried that if I have kids, the world will keep getting worse for them, and I don't want to put them in a bad situation*" (Participant # 116, girl, age 16, city, Alberta).

For several participants, these feelings were accompanied by a sense of guilt. For example, one participant stated, "*the change is making me feel very worried about the future that my kids will live in, and I feel guilty about it*" (Participant # 294, girl, age 18, city, Alberta). Another participant reflected, "*I hate to think that my kids will not be able to enjoy the world the same way that I have, and I sometimes feel guilty for thinking about having kids at all*" (Participant # 24, girl, age 17, small town, Newfoundland and Labrador).

### iii. Impacts on Functioning

This theme consists of 8 codes and accounted for 25 coded responses (6.5%), exploring the participants' perspectives on the impact climate change has had on their physical health and daily functioning. This theme incorporates two subthemes. Impacts on daily functioning were captured when key aspects of an adolescent's daily life were affected by climate change. Physiological impacts included comments on physical health effects.

### iii.i. Impact on Daily Functioning

Participants' perceived impact of climate change on their daily routines appeared in 16 coded responses (4.2%). An aspect of daily functioning that participants reported as influenced by climate change was sleep. For example, one participant expressed that thinking about climate change caused stress, which in turn hindered their sleep: "*it just makes me have a hard time sleeping as I'm thinking about it and makes me stressed*" (Participant #318, boy, age 17, small town, Ontario). Another participant stated, "*it makes me restless at night*" (Participant #427, boy, age 14, city, British Columbia). Some participants reported that, in addition to issues with sleep, concerns about climate change also affected their

concentration at school. As one participant stated, "*keeps me up at night, can't focus on school as good*" (Participant # 370, boy, age 16, small town, British Columbia).

iii.ii. **Physiological Impacts**

Participants' perspectives about how climate change impacts their physical health, including difficulty breathing, head-aches, and weakness, appeared in 9 coded responses (2.4%). Participants expressed how climate change was impact-ing their mental health through physiological systems. For example, one participant shared, "*the different temperatures can make my body weak, and my mind works slowly*" (Participant # 322, girl, age 16, city, Ontario). One participant also shared the consequences of recent wildfires on their physiology, stating "*the fires this summer made it hard to breathe*" (Participant # 496, girl, age 18, small town, Saskatchewan). Generally, participants seemed to make the connection between their physical health and mental health in the context of climate change. For example, a participant expressed how adverse weather conditions impacted both their physical and mental health simultaneously, stating, "*the worse the weather gets, the worse my mental health gets, it causes headaches and more*" (Participant # 725, girl, age 14, city, Ontario).

iv. **Concerns for the Environment, Humanity, and Wildlife**

This theme consists of three codes and accounted for 34 coded responses (8.9%), encompassing responses in which participants expressed concerns about the natural environment, humanity, and wildlife. One participant stated, "*it worries me when I see my surroundings, and then I think of all the things of nature that I'm blessed with might go away due to climate change*" (Participant #57, girl, age 17, city, Alberta). Other participants expressed sadness and distress concerning the adverse impacts of climate change on wildlife. One participant expressed, "*I see all the animals dying or losing their homes*" (Participant # 93, girl, age 15, city, Ontario). Another participant shared, "*it sometimes makes me think about peo-ple in other countries who are innocent, as well as poor animals that have done nothing wrong*" (Participant # 187, boy, age 17, small town, British Columbia).

## 4. Discussion

This study explores if and how Canadian adolescents perceive that climate change is impacting their mental health. In this sample, over one-third of the survey respondents felt climate change was impacting their mental health. Incorporating open-ended questions as follow-ups to broader closed-ended questions provides deeper insights into respondents' expe-riences and perspectives on how this was occurring. Our findings highlight that adolescents' perspectives on how climate change impacts their mental health varied, suggesting that multiple mechanisms may contribute to this connection.

A notable finding is that, although substantial, the proportion of participants reporting that climate change has an impact on their mental health was lower in the present study (36%) compared to similar studies in overlapping age groups. For example, Galway and Field [32] found that almost 78% of surveyed 16–25-year-olds felt that their feelings about climate change was impacting their mental health. Another Canadian survey, the Ontario Student Drug Use and Health Survey, [45] found that approximately 45% of students in grades 6–10, reported feeling depressed about the future due to climate change. While asking about feeling worry rather than overall mental health impacts, Hickman et al.'s [28] survey of 10,000 young people (aged 16–25 years) in ten countries revealed that 59% of the participants felt 'very' or 'extremely' worried about climate change, while 84% were at least moderately worried. Such differences in estimates may reflect a true distinction between the adolescents surveyed in the present study and the overlapping but distinct populations in others studied; however, it also highlights that consideration is needed in terms of when a survey was conducted, in relation to weather and news about extreme events, as well as how the survey questions are worded and employed. For example, Galway and Field asked, "To what degree do your feelings about climate change impact your overall mental health?" and provided five possible response categories ranging from "not at all" to "a very great deal," with 4/5 responses being

affirmative [32]. While the present study asked a similar question ("Do you think climate change is impacting your mental health?"; responses options ranging from 'not at all' to 'a lot', with 2/3 options in the affirmative), although for both surveys this question was positioned after questions about climate emotions, both question wording and response options have been demonstrated to influence participant responses [46,47]; therefore, further research is needed to explore survey design when examining the mental health impacts of climate change among adolescents.

This study highlights several pathways through which climate change is impacting mental health among adolescents, aligning with existing theoretical frameworks that emphasize multiple direct, indirect, and overarching pathways [14]. For example, participants in this study linked wildfires and extreme heat events to negative emotions and psychological responses, such as fear, anxiety, and emotional discomfort, supporting the theory that both awareness and direct experiences of such events can trigger affective responses [48,49]. Further, study participants expressed anger and frustration towards systemic barriers to climate action, reflecting an indirect political pathway as described by Niedzwiedz et al. [14], where inadequate institutional responses amplify distress. This echoes previous findings of concern among adolescents about the impact of worsening climate events and distress and frustration about a lack of climate action [28,50]. Participants in this study also reported mental health impacts of climate change stemming from exposure to climate-related information and a general awareness of climate change. Previous studies have highlighted that awareness and knowledge about the issue of climate change and the threat it poses to the Earth's systems can serve as an overarching factor contributing to negative affect [15–17].

Concerns for the future as sources of negative emotions and psychological responses, like fear, anxiety, and worry, emerged from the participants' responses. Climate change may cause stress for many adolescents, and the unpredictability and severity of it can interfere with daily routines and create feelings of instability [51,52]. Additionally, some participants expressed worry and guilt about considering bringing children into the deteriorating world, aligning with broader societal debates around the morality of having children in the face of ongoing climatic change [53,54].

Participants' responses indicated the impacts of climate change on daily functioning and physical health, such as detrimental effects on sleep and school performance. This finding is supported by previous reviews [55,56]. For example, Burke, Sanson, and Van Hoorn, in their narrative review about the psychological effects of climate change on children, reported that climate change put children at risk of mental health consequences including depression, anxiety, and sleep disorders, which in turn can impact emotion regulation, cognition, behavior, and academic performance. In their review, they outline that climate change threatens children's mental and physical health by destabilizing the socio-economic and environmental determinants necessary for healthy development, including stable families and communities [55,56]. This connected view of health is supported in this study as multiple adolescents described physical health impacts of climate change, such as headaches, body weakness, and respiratory issues, when asked about how climate change is effecting their mental health. Further, our findings supports that physiologic effects are a pathway through which climate change impacts adolescents' mental health, as outlined by Proulx et al [57] in their narrative review about how climate change impacts on child and adolescent health and well-being. As physical and mental health are connected [58]; our findings support looking at adolescent health holistically when considering climate change impacts [14].

## 4.1 Implications

Understanding adolescents' perspectives and lived experiences is critical in identifying and understanding age-specific health implications associated with climate change. Given that a substantial proportion of the study sample felt that climate change was impacting their mental health, and when asked how they experienced this, a range of perceptions and experiences were highlighted, this study indicates that there is a pressing need for increased support for adolescents as they face the consequences of climate change. Additionally, responses from gender-diverse participants and those with existing mental health conditions revealed that there might be some intersecting factors that could put some people at greater risk for the mental health impacts of climate change, which warrant further exploration.

Future research is needed to identify evidenced-based approaches that empower adolescents and provide them with supportive programs to reduce risks to mental health and foster resilience in the face of climate change [59]. Programs could be developed to support adolescents in developing positive coping strategies for the difficult emotional responses they experience due to concerns about climate change. There are some approaches that might be promising, but we caution that future studies are needed to examine their effectiveness among adolescents. For example, promotion and expansion of climate cafes as safe emotional spaces may provide active-listening environments where young people who wish to can share their climate change related emotions without being judged or pressure to 'fix' their feelings [60]. Further, school-based programs that offer young people opportunities to develop a sense of agency and empowerment about climate change in a positive environment [61] may support adolescents' mental wellbeing. Emotional literacy could be encouraged within families and groups that work with young people to support psychological wellbeing when dealing with uncertain outlooks toward the future; this may also create an outlet for sharing and discussing concerns around parenthood [62]. Public health organizations should also acknowledge and plan for the mental health challenges experienced by adolescents that stem from climate change. Additionally, school and community design that aims to mitigate the impact of climate change related events on health, such as shaded outdoor spaces, is an important area for future work.

Anger and frustration regarding the perceived inaction of governments was identified among adolescents in this study. This highlights the need for policymakers and global leaders to recognize the psychological toll of climate inaction on the younger generation and to work with adolescents to develop and communicate climate change policies [63]. Youth engagement in climate change discussions and decision-making processes can empower young people while addressing systemic barriers and their concerns about climate change.

## 4.2 Limitations and strengths of the study

This study used open-ended survey questions which allowed for insights into adolescents' perspectives and experiences regarding how climate change was impacting their mental health in their own words. Another key strength of this paper is the involvement of youth co-researchers, thereby embodying a collaborative approach to conducting research with youth rather than merely on youth.

We would also like to acknowledge some of this study's limitations. First, selection bias could be present in the recruitment process, as participants agreed to participate in the survey with the knowledge that it addressed climate change. This may have led to an overrepresentation of adolescents who are concerned about the topic, potentially skewing the results toward stronger perceptions of its mental health impacts. Second, self-reported responses could introduce desirability reporting and recall bias. Third, Canada's 2023 wildfire season was the most destructive in the country recorded at the time, having consumed 15 million hectares of land [64] and preceded the timing of this survey, resulting in its potential influence on participants' responses. Future longitudinal studies could clarify the association between exposure to such extreme events and mental health impacts, addressing temporal variations. Fourth, while the survey recruited participants from across Canada, it is based on non-probability sampling and therefore not representative of the general population, for example, racialized youth and those with Indigenous identity represented 39% and 2.6% of our sample, respectively. However, according to the Statistics Canada, self-identified racialized youth (aged 15–19 years) account for 31.4% and Indigenous youth 7.4% [65]. Communities that don't speak English, such as Francophone communities (it is estimated that 6.9% of national youth population can speak only French [66]), as well as those without internet access, such as remote communities, are likely not represented, as the survey was conducted in English and online. Future work should validate and implement the survey questions in other languages to gain broader inclusivity. Similarly, in-person or telephone interview approaches could be conducted in regions with limited internet access. Further, although open-ended questions allow for elaboration and description from participants, we were unable to clarify or ask follow-up questions to gain deeper insights. Future research should prioritize in-depth qualitative studies to investigate further the mental health impacts of climate change on adolescents. Such studies could be crucial in shaping effective mitigation strategies.

## 5. Conclusion

This study offers insights into the perspectives of Canadian adolescents on how climate change impacts their mental health. Participants expressed a range of ways in which climate change was impacting their mental health. This occurs through feelings of depression, anxiety, worry, sadness, stress, hopelessness, and guilt, as well as physical health impacts and disruptions in daily functioning. Such experiences were often linked to extreme climate-related weather events and concerns about loved ones, the environment, wildlife, their future and future generations, as well as lack of action by those in power. Policymakers and public health leaders can learn from adolescents' voices and efforts should be made to mitigate climate change impacts on the mental health of present and future generations.

## Supporting information

**S1 Table. Codebook**
(XLSX)

**S2 Table. Summary Table of Code Counts S1 Table**
(XLSX)

## Author contributions

**Conceptualization:** Ishwar Tiwari, Rebekah Alice McKinnon, Azeezah Jafry, Gina Martin.

**Data curation:** Ishwar Tiwari.

**Formal analysis:** Ishwar Tiwari, Rebekah Alice McKinnon, Azeezah Jafry, Ekroop Grewal.

**Funding acquisition:** Gina Martin.

**Investigation:** Ishwar Tiwari, Rebekah Alice McKinnon, Azeezah Jafry.

**Methodology:** Ishwar Tiwari, Rebekah Alice McKinnon, Azeezah Jafry, Ekroop Grewal, Gina Martin.

**Project administration:** Ishwar Tiwari.

**Resources:** Gina Martin.

**Software:** Gina Martin.

**Supervision:** Gina Martin.

**Writing – original draft:** Ishwar Tiwari.

**Writing – review & editing:** Rebekah Alice McKinnon, Azeezah Jafry, Ekroop Grewal, Jason Gilliland, Kendra Nelson Ferguson, Kiffer G. Card, Maya Gislason, Alina Paula Cosma, Gina Martin.

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
