## [Decision Letter · Decision Letter 0]

25 Jun 2025

PMEN-D-25-00221

Canadian Adolescents’ Perceptions of how Climate Change is Impacting their Mental Health: A Qualitative, Cross-sectional Analysis

PLOS Mental Health

Dear Dr. Martin,

Thank you for submitting your manuscript to PLOS Mental Health. After careful consideration, we feel that it has merit but does not fully meet PLOS Mental Health’s publication criteria as it currently stands. Therefore, we invite you to submit a revised version of the manuscript that addresses the points raised during the review process.

Thank you for your submission. Two of three reviewer suggested minor changes, one reviewer suggested a major revision. Please read the comments carefully and resubmit a revised version of your manuscript.

We look forward to receiving your revised manuscript.

Kind regards,

Paolo Raile

Academic Editor

PLOS Mental Health

Journal Requirements:

1. Please ensure that your Ethics Statement is available in its entirety at the beginning of your Methods section, under a subheading 'Ethics Statement'.

2. For studies involving third-party data, we encourage authors to share any data specific to their analyses that they can legally distribute. PLOS recognizes, however, that authors may be using third-party data they do not have the rights to share. When third-party data cannot be publicly shared, authors must provide all information necessary for interested researchers to apply to gain access to the data. (https://journals.plos.org/plosone/s/data-availability#loc-acceptable-data-access-restrictions) 

Additional Editor Comments (if provided):

Reviewers' comments:

Reviewer's Responses to Questions

**Comments to the Author**

1. Does this manuscript meet PLOS Mental Health’s publication criteria?

Reviewer #1: Yes

Reviewer #2: Yes

Reviewer #3: Yes

2. Has the statistical analysis been performed appropriately and rigorously?

Reviewer #1: Yes

Reviewer #2: Yes

Reviewer #3: N/A

3. Have the authors made all data underlying the findings in their manuscript fully available (please refer to the Data Availability Statement at the start of the manuscript PDF file)?

Reviewer #1: Yes

Reviewer #2: No

Reviewer #3: Yes

4. Is the manuscript presented in an intelligible fashion and written in standard English?

Reviewer #1: Yes

Reviewer #2: Yes

Reviewer #3: Yes

Reviewer #1: Feedback on Manuscript: Canadian Adolescents' Perceptions of how Climate Change is Impacting their Mental Health: A Qualitative, Cross-sectional Analysis

General Comments:

This manuscript presents a well-conducted qualitative study exploring Canadian adolescents’ perceptions of how climate change impacts their mental health. The topic is timely and significant, given the growing recognition of climate change as a determinant of mental health, particularly among youth. The use of open-ended survey responses to capture nuanced perspectives is a strength, as is the involvement of youth co-researchers, which enhances the study’s authenticity and relevance. The thematic analysis is robust, and the findings align with existing theoretical frameworks while offering new insights into adolescents’ lived experiences. However, there are areas where clarity, methodological rigor, and discussion could be strengthened to enhance the manuscript’s impact. Below, I provide detailed feedback organized by section, followed by specific recommendations for improvement.

Strengths:

1. Relevance and Originality: The study addresses a critical gap in the literature by focusing on adolescents’ qualitative perspectives on climate change’s mental health impacts, an area underexplored compared to quantitative studies. The findings contribute to understanding the diverse pathways (direct, indirect, and overarching) through which climate change affects youth mental health.

2. Youth Involvement: The inclusion of youth co-researchers in survey design and coding is a significant strength, aligning with participatory research principles and ensuring the study reflects the voices of the target population.

3. Thematic Analysis: The inductive thematic analysis with semantic coding is well-executed, with clear descriptions of the coding process, including the use of multiple coders and consensus-building to ensure rigor. The four identified themes (emotional/psychological responses, concerns for the future, impacts on functioning, and concerns for the environment/humanity/wildlife) are well-supported with participant quotes.

4. Contextual Relevance: The study is grounded in the Canadian context, referencing specific climate-related events (e.g., wildfires, heatwaves) that resonate with participants’ experiences, making the findings relevant to policymakers and public health practitioners in Canada.

5. Ethical Considerations: The manuscript clearly outlines ethical approvals, participant consent processes, and data anonymity, adhering to high ethical standards.

Areas for Improvement:

1. Introduction:

o Clarity of Research Gap: While the introduction highlights the lack of qualitative studies on adolescents’ perspectives, it could more explicitly articulate how this study builds on or differs from existing qualitative work (e.g., Galway & Field, 2023). A brief comparison with prior qualitative studies globally could strengthen the justification for the study.

o Theoretical Framework: The introduction mentions direct, indirect, and overarching pathways (citing Niedzwiedz et al., 2024) but does not sufficiently elaborate on how these pathways frame the study’s objectives or analysis. A clearer integration of this framework in the introduction would provide a stronger theoretical grounding.

2. Methods:

o Sampling and Selection Bias: The manuscript acknowledges potential selection bias due to participants’ prior knowledge of the survey’s focus on climate change. However, it does not discuss how this bias might have influenced the findings or how the recruitment through Qualtrics’ pre-registered panel may have shaped the sample (e.g., potential overrepresentation of engaged or environmentally conscious youth). More discussion on the panel’s characteristics and recruitment process would enhance transparency.

o Survey Design: The description of the survey instrument development is robust, but the manuscript could provide more detail on the specific climate-related questions asked before the open-ended prompt to contextualize participants’ responses. For example, how were “climate emotions” questions framed, and could they have primed certain responses?

o Data Exclusion Criteria: The exclusion of 56 responses (42 deemed too vague or irrelevant, 14 missing) is noted, but the manuscript lacks a clear explanation of the criteria used to determine vagueness or irrelevance. Providing examples of excluded responses or a more detailed justification would improve methodological transparency.

o Sample Representativeness: The study aimed for equal gender and age representation, which is commendable, but the underrepresentation of Francophone and non-English-speaking communities is a significant limitation. The manuscript could discuss strategies to address this in future research, such as multilingual surveys or targeted recruitment in Francophone regions.

3. Results:

o Theme Descriptions: The four themes are well-defined, but some subthemes (e.g., “negative emotions about climate change connected to climate change information”) have relatively few coded responses (n=5). The manuscript could justify why these were retained as distinct subthemes rather than merged with broader categories (e.g., general affective state).

o Quantitative Integration: The inclusion of code frequencies (Table 2) is valuable, but the manuscript could better integrate these quantitative findings with the qualitative themes. For example, discussing why certain themes (e.g., general affective state, n=119) were more prevalent could provide deeper insights.

o Participant Context: While quotes include age and gender, adding brief contextual details (e.g., urban/rural residence or region) could enrich the interpretation of responses, especially given Canada’s geographic diversity and varying exposure to climate events.

4. Discussion:

o Comparison with Existing Literature: The discussion compares the study’s findings to Galway and Field (2023) and the Ontario Student Drug Use and Health Survey (2024) but could engage more deeply with global studies (e.g., Hickman et al., 2021) to highlight how Canadian adolescents’ experiences align with or differ from international findings.

o Implications: The implications section is strong in advocating for youth-focused interventions (e.g., climate cafes, policy engagement), but it could be more specific about how these interventions might address the identified themes. For example, how could climate cafes mitigate concerns about parenthood or systemic barriers?

o Limitations: The limitations section acknowledges selection bias and language barriers but could further discuss the cross-sectional design’s inability to capture temporal changes in perceptions or the potential influence of recent climate events (e.g., 2023 wildfires) on responses.

5. Clarity and Structure:

o Terminology Consistency: Terms like “climate distress” and “climate anxiety” are used interchangeably in the introduction but not clearly defined. A brief definition or consistent use of terminology would improve clarity.

o Table 1: The table is informative but contains minor inconsistencies (e.g., “Well-off” is listed as 33% for the subset in the table but 27.2% for the full sample). Double-checking these figures for accuracy is recommended.

o Word Count: The body word count (6152) is substantial for a qualitative study. Consider condensing repetitive sections (e.g., some participant quotes could be shortened or summarized) to improve readability without losing meaning.

Specific Recommendations:

1. Introduction:

o Add a paragraph or sentence comparing this study to existing qualitative work globally to clarify its unique contribution.

o Explicitly link the theoretical framework (direct, indirect, overarching pathways) to the study’s research questions and analysis plan.

2. Methods:

o Provide more detail on the Qualtrics panel recruitment process, including how participants were selected and any incentives offered beyond “compensation per panel terms.”

o Clarify the criteria for excluding vague or irrelevant responses, potentially including examples in the supplementary material.

o Discuss strategies to address the underrepresentation of Francophone and non-English-speaking communities in future research.

3. Results:

o Justify the retention of low-frequency subthemes or consider merging them with broader categories to streamline the findings.

o Include additional participant context (e.g., region or urban/rural status) with quotes to reflect Canada’s geographic diversity.

o Integrate code frequencies more explicitly in the narrative to highlight the relative prominence of themes.

4. Discussion:

o Expand the comparison with global studies to situate the findings in a broader context.

o Provide more specific recommendations for interventions tailored to the identified themes (e.g., addressing guilt about parenthood or systemic barriers).

o Acknowledge the potential influence of recent climate events (e.g., 2023 wildfires) on participants’ responses and discuss how longitudinal studies could address temporal variations.

5. Clarity and Presentation:

o Define key terms (e.g., climate anxiety, climate distress) early in the manuscript and use them consistently.

o Verify the accuracy of Table 1 percentages and ensure consistency across the text and tables.

o Condense the manuscript by summarizing repetitive quotes or sections to reduce the word count to approximately 5000–5500 words for better readability.

6. Supplementary Material:

o The supplementary material is referenced but not described in detail. Provide a brief overview of its contents (e.g., list of 85 subcodes) in the main text or ensure it is accessible for reviewers to assess its relevance.

Overall Assessment:

This manuscript makes a valuable contribution to the literature on climate change and adolescent mental health, particularly by amplifying youth voices through qualitative methods. The thematic analysis is rigorous, and the findings offer actionable insights for policymakers and public health practitioners. However, addressing the identified areas for improvement particularly clarifying the research gap, enhancing methodological transparency, and strengthening the discussion’s global context will elevate the manuscript’s impact and suitability for publication in PLOS Mental Health. I recommend major revisions to address these points, with a focus on streamlining the manuscript and enhancing its theoretical and methodological clarity.

Recommendation: Major Revisions

Reviewer #2: This article, “Canadian Adolescents’ Perceptions of how Climate Change is Impacting their Mental Health: A Qualitative, Cross-sectional Analysis”, makes a timely and meaningful contribution to the intersection of adolescent mental health and climate science. The use of inductive thematic analysis on open-ended responses offers rare and valuable insights into how Canadian adolescents personally experience climate-related emotional burdens. The authors have done commendable work in elevating youth voices and presenting a nuanced picture of emotional, existential, and functional impacts.

Strengths:

The methodological design is robust, with clear justification for qualitative approaches and quota-based sampling ensuring geographic, gender, and age diversity.

The involvement of youth co-researchers throughout the process enhances the study’s credibility and relevance.

The five subthemes within emotional and psychological responses are particularly well-synthesized and illustrative.

Recommendations:

Clarify demographic limitations: While the paper mentions quota fulfillment, it would benefit from more detail on how representative the sample is of racialized youth, those from Francophone and Indigenous communities, and adolescents without internet access.

Deepen intersectional analysis: Some respondents’ quotes suggest layered vulnerabilities (e.g. gender-diverse youth, those with pre-existing mental health conditions), but this nuance could be explored further in discussion.

Actionable implications: The “Implications” section could be expanded to include more specific, youth-informed policy recommendations or examples of promising practices (e.g. school-based supports, community resilience programs).

Thematic contextualization: A table cross-referencing themes with demographic factors (e.g. age or region) could add further depth and help readers understand if different impacts are more common among certain subgroups.

Overall, this manuscript is a well-structured and thoughtful addition to the literature on climate mental health and youth. With some refinements, it has potential for strong impact in both academic and policy-making spheres.

Reviewer #3: Thank you for the opportunity to review this interesting article.

The article used open-ended responses to a question in a cross-sectional survey to assess how climate change was impacting Canadian’s adolescents.

The article addressed an important question. Collection of primary data, specifically, a large nationally representative sample, from Qualtrics is a strength. The analyses are thorough, with the involvement of youth co-researcher a particular strength. The findings are informative. The paper is generally well written, giving voices to a largely neglected yet very important population in this sphere.

Below are my minor suggestions to further improve the paper:

1) Qualitative Methodology: Perhaps it is my own limitation, but cross-sectional survey is usually used to describe quantitative research design. Qualitative research usually includes methods like interviews or field observations. The combination of “qualitative, cross-sectional” comes across a bit odd or awkward. Perhaps you can simply refer to it as an open-ended survey.

In addition, it would be helpful to clarify your epistemological stance, whether it is positivist or interpretivist.

2) Participant: Please clarify, at the very beginning of the Methods section, what is your population of interest. Inclusion and exclusion criteria should be listed before description of any procedures.

A phrase is useful to concisely describe whether you use convenience or probability sampling and whether the stratification aims to ensure national representativeness.

A response rate should be provided. Non-responses should be characterized and compared to responses in terms of sociodemographic characteristics to address response and sample selection biases.

3) Measure: You seem to have used different tenses interchangeably throughout the paper. Your question used present tense “is impacting.” In alignment with that, you should use “was impacting” rather than “impacted” or “impact.” This is not merely language policing, but this focus on the “present” should somehow reflect in your theorization of the constructs and your interpretation of the findings.

4) Limitations: The open-ended responses may still suffer from common measurement errors of self-reports, including social desirability and recall biases, which should be acknowledged as a limitation.

Congratulations on the impressive project! Best luck moving forward!

**Do you want your identity to be public for this peer review?** For information about this choice, including consent withdrawal, please see our Privacy Policy

Reviewer #1: **Yes: ** Neelam Punjani

Reviewer #2: **Yes: ** Vitalii Klymchuk

Reviewer #3: No

---

## [Decision Letter · Decision Letter 1]

14 Aug 2025

Canadian Adolescents’ Perceptions of how Climate Change is Impacting their Mental Health: A Qualitative Analysis of Open-Ended Survey Responses

PMEN-D-25-00221R1

Dear Dr Martin,

We are pleased to inform you that your manuscript 'Canadian Adolescents’ Perceptions of how Climate Change is Impacting their Mental Health: A Qualitative Analysis of Open-Ended Survey Responses' has been provisionally accepted for publication in PLOS Mental Health.

Best regards,

Paolo Raile

Academic Editor

PLOS Mental Health

Thank you for your submission. I am pleased to inform you that your article has been accepted.

Reviewer Comments (if any, and for reference):

Reviewer's Responses to Questions

**Comments to the Author**

Reviewer #2: All comments have been addressed

Reviewer #3: All comments have been addressed

publication criteria?

Reviewer #2: Yes

Reviewer #3: Yes

3. Has the statistical analysis been performed appropriately and rigorously?

Reviewer #2: Yes

Reviewer #3: Yes

4. Have the authors made all data underlying the findings in their manuscript fully available (please refer to the Data Availability Statement at the start of the manuscript PDF file)?

Reviewer #2: Yes

Reviewer #3: Yes

5. Is the manuscript presented in an intelligible fashion and written in standard English?

Reviewer #2: Yes

Reviewer #3: Yes

Reviewer #2: No further comments

Reviewer #3: Thank you for addressing my comments.

**Do you want your identity to be public for this peer review?** For information about this choice, including consent withdrawal, please see our Privacy Policy

Reviewer #2: No

Reviewer #3: No
